# ROBUST SCHEDULING WITH GFLOWNETS

**David W. Zhang**[*]
University of Amsterdam

**Corrado Rainone   Markus Peschl   Roberto Bondesan**[†]
Qualcomm AI Research

## ABSTRACT

Finding the best way to schedule operations in a computation graph is a classical NP-hard problem which is central to compiler optimization. However, evaluating the goodness of a schedule on the target hardware can be very time-consuming. Traditional approaches as well as previous machine learning ones typically optimize proxy metrics, which are fast to evaluate but can lead to bad schedules when tested on the target hardware. In this work, we propose a new approach to scheduling by sampling proportionally to the proxy metric using a novel GFlowNet method. We introduce a technique to control the trade-off between diversity and goodness of the proposed schedules at inference time and demonstrate empirically that the pure optimization baselines can lead to subpar performance with respect to our approach when tested on a target model. Furthermore, we show that conditioning the GFlowNet on the computation graph enables generalization to unseen scheduling problems for both synthetic and real-world compiler datasets.

## 1   INTRODUCTION

Efficient execution of computation graphs is paramount to many scientific and industrial applications, with deep learning being a prominent example (Amodei & Hernandez, 2018). Scheduling is the action of assigning operations to the available compute resources, such as threads, cores, or nodes in a cluster (Kwok & Ahmad, 1999; Hennessy & Patterson, 2011; Pinedo, 2012). Unfortunately, finding the schedule with the shortest possible *makespan* (start-to-end runtime) is in general NP-hard (Papadimitriou & Steiglitz, 1998). As a result, domain experts have come up with heuristics that are tailored to specific problem instances (Ibarra & Kim, 1977). Machine learning approaches promise the possibility to automate this process allowing for fast adaptation to new graph distributions (Wang & O'Boyle, 2018; Bengio et al., 2021c). In this work, we consider the problem of scheduling a set of operations with precedence constraints on a fixed number of homogeneous devices, i.e., any operation can run on any device and the runtime is the same on all devices.

Evaluating the makespan of a schedule involves running all operations in the computation graph on some target hardware. This can be very resource intensive, especially when the computation graph includes lengthy operations, the evaluated schedule is inefficient, or the intended target hardware is a cluster with many nodes. Heuristic optimizers, like genetic algorithms (Hou et al., 1994), or machine learning (Mao et al., 2019) approaches further exacerbate this problem because they require many evaluations to converge (Chen et al., 2018). Proxies are a much faster alternative that estimates the makespan using a simplified model of the hardware. However, this comes at the cost of discrepancies between the proxy makespan and the one observed on the hardware; as a result, performant solutions on the proxy might ultimately be unsatisfactory once tested on the target. Nonetheless, proxies remain a good indicator for most schedules and are essential due to their efficiency. We aim to learn a scheduler that can be trained using the proxy, whilst being robust to its inaccuracies.

The common approach to scheduling problems (and combinatorial optimization problems in general) is to look for the single best schedule that minimizes a makespan measure which can be an analytical proxy (Paliwal et al., 2020), the output of a simulator (Zhou et al., 2020), or even the real makespan on hardware (Khadka et al., 2021). We propose a different philosophy: generate a set of candidate schedules that have a low makespan according to the proxy and are diverse. By hav-

---

[*]Work completed during internship at Qualcomm Technologies Netherlands B.V.; Qualcomm AI Research is an initiative of Qualcomm Technologies, Inc.
  [†]This work was done while the author worked at Qualcomm Technologies, Inc.

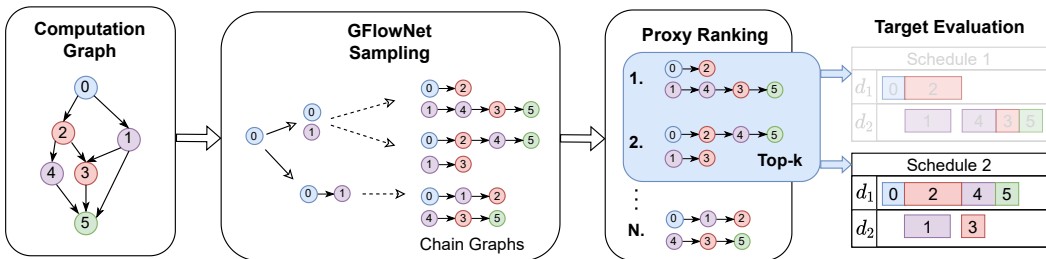

Figure 1: Full pipeline of our generative scheduling approach. Conditioned on the computation graph we generate multiple candidate schedules using GFlowNet, filter for the best $k$ with the proxy and pick the best performing one out of the $k$ that we check on the target. Here we illustrate the pipeline for $k = 2$ and two devices, $d_1, d_2$.

ing multiple good schedules that are significantly different, we can reduce the impact of systematic errors in the proxy, and hope for robust performance on the target.

Our goal is to learn a generative model that assigns higher probability to low-makespan schedules, and importantly can also discover the different modes associated with local optima of the makespan cost. Generative Flow Networks (GFlowNets) have recently been introduced as a method for learning a stochastic policy that can piece-by-piece construct discrete and composite objects, proportional to a given reward (Bengio et al., 2021b). By computing the reward from the proxy-makespan we can use GFlowNets to sample a diverse set of candidate schedules.

**Our main contributions are: 1.** We introduce an alternative to the pure proxy optimization viewpoint of scheduling that achieves better robustness to proxy errors, by generating multiple candidate schedules to evaluate directly on the target hardware. **2.** We extend GFlowNets to generate schedules conditioned on a computation graph. Additionally, we introduce a method to control diversity and goodness at inference time, without the need for retraining. These contributions may be of general interest, beyond the scheduling problem. **3.** We empirically demonstrate the robustness of our method to proxy errors and verify the generalization ability on a diverse set of synthetic and real-world computation graphs.

## 2 ROBUST SCHEDULING

In this section, we first provide a definition of the scheduling problem we consider in this work. Then, we discuss how a proxy simulates the schedule execution as well as the difficulties of specifying a reliable proxy. Finally, we describe our proposed generative scheduling framework.

### 2.1 PROBLEM DEFINITION

In scheduling, we are given a computation graph $G_C = (O, P)$ that is a direct acyclic graph (DAG) consisting of operations (nodes) $o \in O$ and precedence constraints (edges) $p \in P$ that encode a partial order in which the operations need to be executed. In particular, the edge $p_{ij}$ encodes that operation $o_i$ needs to finish before $o_j$ can start, for example because $o_j$ requires the output of $o_i$ as input. Our task is to run all operations on a set of *devices* $\mathcal{D} = \{d_1, \ldots, d_m\}$, without violating the precedence constraints. In addition to the precedence constraints, the devices can only run one operation at a time. We can then view scheduling as performing two distinct tasks: assign a device to each operation, and determine a (complete) order among all operations on the same device that is compatible with the precedence constraints encoded in $G_C$. We can model the schedule as a chain of operations for each device, where the chain denotes the order in which the operations run on that device. See Figure 1 for a visual example of the chain graphs. Our aim is to find the schedule with the lowest makespan for some target hardware.

### 2.2 TARGET MODEL VS. PROXIES

The makespan of any schedule can be evaluated on the target hardware by running all the operations in the specified order and on the specified devices. However, this can take up significant time

and compute resources when the computation graph is large, has costly operations, or the target hardware is a cluster with many nodes. In addition to this, when optimizing the makespan one needs to evaluate many different schedules, further exacerbating the resource requirements.

A *proxy* is any tool that allows one to estimate the makespan of a given schedule, without having to run the schedule on the target hardware. Proxies come with significant speed advantages, which remedy the problems mentioned above. However, this comes at the cost of possible mistakes in the estimation of the makespan and relative comparison of schedules. Mistakes can occur for example when the task durations are not accurately profiled, memory movements are too complex to fully model, or additional hardware-specific features are changed. Ideally, we would like to rely on a proxy for the majority of the schedule evaluations, and only evaluate a small fraction of promising schedules on the target hardware. This approach differs from previous works, that either evaluate every schedule on the target (Khadka et al., 2021), leading to very long optimization times, or evaluate everything on the proxy (Paliwal et al., 2020), which is susceptible to modeling failures.

Next, we describe how the proxy we use in this work assigns start times to each operation given a schedule and estimates the makespan based on those. We recall that a schedule is an order of operations for each device, which can be represented by one chain graph per device. For each of these, let us denote with $C_d$, $d \in \mathcal{D}$ the set of edges of the chain graph for device $d$ and with $D := \bigcup_{k=1}^{m} C_{d_k}$ the set of all device constraints. The operations correspond to graph nodes and are labeled in the same way as in $G_C$. No other operation can run on the same device during the *runtime* or *duration* $\rho_i$ of operation $o_i$ In practice, $\rho_i$ is estimated directly on the hardware in a profiling stage that precedes scheduling. We denote the start time of $o_i$ as $\tau_i$ and can thus express the precedence constraints as:

$$\tau_j \geq \tau_i + \rho_i, \quad \forall (i, j) \in P \cup D \tag{1}$$

An operation cannot start unless all of those that produce its inputs and all of those that precede it on its assigned device have finished first. To ensure that these constraints are satisfied the proxy assigns each operation $o_i$ the start time

$$\tau_i = \max_k \{\tau_k + \rho_k | (k, i) \in P \cup D\} \tag{2}$$

If a node has no parents in $P \cup D$ the proxy assigns the start time $\tau_i = 0$. The start times of all $o_i \in O$ can be computed by assigning a start time to a node whenever it has no parents or all its parents have an assigned start time. If the graph $(O, P \cup D)$ is a DAG, then this algorithm is guaranteed to assign start times that satisfy Equation 2. The proxy then estimates the *makespan* $T$ of the schedule $x$ as:

$$T(x) := \max_i (\tau_i + \rho_i) - \min_i (\tau_i) \tag{3}$$

Optimizing this cost over the set of all possible schedules is already a very rich problem, and yet, we made significant simplifying assumptions in the construction of the proxy. In particular, we assume perfectly estimated runtimes, and in Equation 2 we effectively assume that an operation can start as soon as all of the operations producing its inputs finish, meaning that data can be moved between devices instantaneously (zero latency) independently of their size (infinite bandwidth). These assumptions might be unrealistic, depending on the specific target devices (Valiant, 1990).

## 2.3 GENERATIVE SCHEDULING

Our aim is to come up with a good set of candidate schedules to be tested on the target hardware while relying only on the proxy for generating this set. While the proxy is imperfect, it still offers good guidance for most schedules; thus, we would like to include schedules that perform well according to the proxy. Nevertheless, we also know that systematic errors in the proxy can cause it to incorrectly predict a low makespan for some schedules. Therefore, we would like the set of candidate schedules to be diverse, while still high-quality from the point of view of the proxy.

If we had a ranking over all the schedules according to the proxy, we could just go through the list top-to-bottom, and add a schedule to the batch whenever it is significantly different from the previous ones. A full ranking like this is infeasible to construct, but we can instead learn a generative model that samples higher ranked schedules with higher probability. When generating schedules we need to satisfy the precedence and device constraint outlined in Section 2.1. To avoid generating invalid schedules we construct a schedule in a step-by-step process: start with an empty schedule at the initial state $s_0$, and at each step add an operation to the partial schedule until the schedule contains

all operations at the terminal state $s_n$. At each intermediate state $s_t$, an action $a_t$ consists in picking an operation and assigning it to one of the devices, leading to a new state $s_{t+1}$. We define the set of valid actions at every step $t$ in a way such that the precedence constraints are automatically satisfied. In particular, adding an operation $o_t$ is a valid action if and only if $\forall k : (k, t) \in P$, $o_k$ is already in the partial schedule at state $s_t$. This is a sufficient condition for the final "schedule" graph $(O, P \cup D)$ to be a DAG, implying that the constructed schedule is feasible. The final states represent full schedules $x$, for which we can compute the makespan $T(x)$ with the proxy, given the runtimes $\{\rho_i\}_{i=1}^n$. We compute the relative *speedup* compared to the makespan on a single device as $U(x) = \sum_i \rho_i / T(x)$, from which we compute the reward as we present in the next section.

## 3 GENERATIVE FLOW NETWORKS FOR SCHEDULING

GFlowNets (Bengio et al., 2021a;b) are methods for training a stochastic policy to sample discrete and composite objects proportionally to a given reward. Each object $x$ is generated incrementally by a sequence of actions. In the previous section, we discussed how to limit the action space to guarantee that we sample valid schedules. After a brief introduction to GFlowNets, the following sections will present our proposed extensions that include a new training objective that is suitable for learning conditional GFlowNets and a method for controlling the selectivity of samples at inference time.

### 3.1 BACKGROUND

We start by introducing some notation. We denote by $\boldsymbol{s} = (s_0, s_1, \dots, s_n)$ a *trajectory* that consists of a sequence of states $s_t$. In the case of scheduling, trajectories start with the empty schedule $s_0$, followed by partial schedules, and end with a complete schedule $s_n$. We denote by $\mathcal{T}$ the set of all such trajectories and by $\mathcal{T}_x$ the set of trajectories that end at $x$. Based on this, we define a flow function $F : \mathcal{T} \rightarrow \mathbb{R}^+$ and its associated normalized probability distribution $P(\boldsymbol{s}) = F(\boldsymbol{s})/Z$, $Z = \sum_{\boldsymbol{s} \in \mathcal{T}} F(\boldsymbol{s})$. A flow function that fulfills the condition: $R(x) = \sum_{\boldsymbol{s} \in \mathcal{T}_x} F(\boldsymbol{s})$ (every terminal state has a total flow matching its reward), results in a probability over schedules $P(x) = \sum_{\boldsymbol{s} \in \mathcal{T}_x} F(\boldsymbol{s})/Z$ that is proportional to the reward $P(x) \propto R(x)$, and further entails that $Z = \sum_x R(x)$.

For any Markovian flow, we can decompose the probability of a trajectory in terms of the forward probability:

$$P(\boldsymbol{s}) = \prod_{t=1}^n P_F(s_t|s_{t-1}) \tag{4}$$

This way, we can generate trajectories $\boldsymbol{s}$ by sampling a sequence of actions starting from $s_0$. In Section 2.3 we described how to limit the action space appropriately to guarantee that every sampled schedule is valid. Similarly, we can define a backward probability $P_B$ that factorizes the trajectory probability conditioned on a terminal state:

$$P(\boldsymbol{s}|s_n = x) = \prod_{t=1}^n P_B(s_{t-1}|s_t) \tag{5}$$

The training objectives considered in previous works aim to achieve a consistent flow (Bengio et al., 2021b; Malkin et al., 2022), where consistency means that the flow estimated for the forward direction should equal the flow for the backward direction. A consistent flow $F(\boldsymbol{s})$ for trajectories $\boldsymbol{s} \in \mathcal{T}_x$ can then be written in terms of $P_F$ and $P_B$ and has to fulfill the equality:

$$Z \prod_{t=1}^n P_F(s_t|s_{t-1}) = R(x) \prod_{t=1}^n P_B(s_{t-1}|s_t) \tag{6}$$

Based on this equation, Malkin et al. (2022) propose to estimate $Z$, $P_F$, and $P_B$ by optimizing the *trajectory balance* loss which is the squared difference between the logarithms of the l.h.s. and the r.h.s. of Equation 6.

### 3.2 LOG-PARTITION VARIANCE LOSS

In order to apply the trajectory balance loss in the conditional case, we would need to learn an additional regression model that estimates the log-partition function $\log Z$ conditioned on $G_C$. Training

such a network accurately is difficult but crucial for learning the probabilities $P_F$. In particular, a wrong estimation of $\log Z$ can incorrectly change the direction of the gradients of the loss function. We explain why this occurs in Appendix B. In practice, we found this approach to perform poorly when different computation graphs had large differences in their $\log Z$ value. Instead, we can rewrite Equation 6 to implicitly estimate $\log Z$ based on the forward and backward flows of a single trajectory $s$, where $P_F$ and $P_B$ are neural networks with parameters $\boldsymbol{\theta}$:

$$\zeta(\boldsymbol{s}; \boldsymbol{\theta}) = \log R(x) + \sum_{t=1}^{n} \log P_B(s_{t-1}|s_t; \boldsymbol{\theta}) - \sum_{t=1}^{n} \log P_F(s_t|s_{t-1}; \boldsymbol{\theta}) \tag{7}$$

In the optimal case, $\zeta(\boldsymbol{s}; \boldsymbol{\theta})$ is equal to the true $\log Z$ which is the same for all trajectories corresponding to the same computation graph $G_C$. Thus, our optimization goal turns into minimizing the variance of $\zeta(\boldsymbol{s}; \boldsymbol{\theta})$ over different trajectories $s$ with the loss

$$\mathcal{L}_{\mathrm{V}}(\boldsymbol{s}; \boldsymbol{\theta}) = (\zeta(\boldsymbol{s}; \boldsymbol{\theta}) - \mathbb{E}_{\boldsymbol{s}}\left[\zeta(\boldsymbol{s}; \boldsymbol{\theta})\right])^2 \tag{8}$$

In practice, we use the training distribution to estimate $\mathbb{E}_{\boldsymbol{s}}\left[\zeta(\boldsymbol{s})\right]$ with a mini-batch of sampled trajectories. For more details on the training process, we refer to Appendix C.

We note that by optimizing the log-partition variance loss in Equation 8, one only needs to parametrize the forward and backward probabilities $P_F$ and $P_B$. This is similar to the non-forward trajectory loss mentioned in the appendix by Malkin et al. (2022), which also does not involve learning any state flows, including the initial flow $Z$. However, our loss does not mix forward and backward steps from different trajectories and directly optimizes the consistency of the total flow $Z$ for each trajectory associated with a given computation graph $G_C$.

### 3.3 Temperature-conditioned Topoformer

**Reward temperature.** We compute the reward as a function of the speedup. In particular, we choose $\log R(x; m, \sigma) = (U(x) - m)/\sigma$ where $U(x)$ is the speedup of the schedule $x$, $m$ is the number of devices, and $\sigma \in \mathbb{R}^+$ plays the role of a temperature. The temperature allows us to concentrate the distribution on the modes and control the selectivity of the generator. This is useful since there can be many more schedules with low speedup when compared to good ones. For example, when simply setting the reward equal to the speedup, we observed that finding schedules with high speedup requires a prohibitively large amount of samples. We expect this temperature term to allow trade-offs between diversity and shifting the mean of the sampling distribution towards better schedules.

Previous works on GFlowNets apply a constant temperature value during training and at inference time (Bengio et al., 2021a; Jain et al., 2022). This can lead to low performance (when set too high), and low diversity or unstable training (when set too low). Furthermore, different computation graphs can have different ideal temperature values, making this approach less suitable when learning conditional GFlowNets. Instead, we propose to learn a single model for multiple different reward functions $R(x; m, \sigma)$, by conditioning the policy networks ($P_F$ and $P_B$) on the temperature $\sigma$. Approximating the temperature-conditioned policy with a neural network is feasible because flows for a given temperature can be continuously morphed into flows for any other temperature. Since our reward $R(x; m, \sigma)$ is continuous with respect to the temperature $\sigma$, we expect the change of flow for different temperatures to be learnable by a neural network. We provide a proof for the following theorem in Appendix A.

**Theorem 1** (Flow Continuity). *Let $\{R_i\}_{i=1}^{\infty}$ be a sequence of non-negative reward functions such that for all terminal states $x$, $R_i(x) \to R(x)$ as $i \to \infty$. Then, for any flow $F^R$ with reward $R$, there exists a sequence of flow functions $\{F^{R_i}\}_{i=1}^{\infty}$ with $F^{R_i}(\boldsymbol{s}) \to F^R(\boldsymbol{s})$ for all $\boldsymbol{s} \in \mathcal{T}$.*

The output policy changes more rapidly as a function of the temperature for values close to 0 than for larger values. To account for this, we use the logarithm of the temperature as input to the policy instead. During training, we sample temperatures from the log-uniform distribution with support between $[\log \sigma_{min}, \log \sigma_{max}]$, where $\sigma_{min}$ is a minimum temperature that is necessary for numerical stability. In comparison to sampling from $\mathcal{U}(\sigma_{min}, \sigma_{max})$, this avoids oversampling from high temperature regions that have little difference in the resulting flow network. At inference time, we choose how close the samples are to the mode by adjusting the $\sigma$.

**Topoformer architecture.**  For the neural network architecture of our policy, we use the Topoformer (Gagrani et al., 2022), which has been recently introduced for learning topological orderings of computation graphs. It builds on the Transformer encoder (Vaswani et al., 2017) and additionally masks the multi-head attention depending on the topology of the computation graph. Both forward and backward policies use separate MLP heads on top of a shared Topoformer encoder. Taking inspiration from the successful use of time conditioning in diffusion models (Song et al., 2020; Ho et al., 2020), we add temperature conditioning by first embedding the temperature using an MLP to produce $e_\sigma$, and then reuse the embedding in every first linear layer block of the Topoformer:

$$\texttt{lin}(h, e_\sigma) = \texttt{lin}_{\text{scale}}(e_\sigma) \odot \texttt{lin}(h) + \texttt{lin}_{\text{shift}}(e_\sigma) \tag{9}$$

Here $\texttt{lin}_{\text{scale}}$ and $\texttt{lin}_{\text{shift}}$ are linear layers and $\odot$ is the elementwise multiplication (Perez et al., 2018). In contrast to diffusion models, we observe better performance on large temperature ranges with the ReLU (Nair & Hinton, 2010) activation function. We hypothesize that this is connected to the monotonicity of the underlying policy function with respect to decreasing temperatures (see Corollary 1 in the Appendix) and the propensity for linear extrapolation of ReLU MLPs (Xu et al., 2020). For a detailed description of the neural network architecture, we refer to Appendix D.

**Sub-graph training.**  Training with a full computation graph might not always be necessary and we hypothesize that learning on sub-graphs can lead to policies that generalize to the full computation graph. This can be seen as a form of data augmentation and increases the amount of training data, while simultaneously improving the training time. We shall use sub-graph training for the larger graphs that we study in this work.

## 4  RELATED WORK

**Reinforcement learning for scheduling.**  Reinforcement learning has been the predominant machine learning approach to optimize the makespan for computation graph schedules (Addanki et al., 2019; Paliwal et al., 2020; Zhang et al., 2020). The rewards used include simple analytical proxies of the makespan (Paliwal et al., 2020; Zhou et al., 2020), but also more refined proxies which incorporate modeling of memory movements (Addanki et al., 2019). Khadka et al. (2021) directly train on the target hardware, but consider only a few computation graphs, and do not show generalization to unseen ones. Addanki et al. (2019) use a sophisticated simulator of the makespan which is customized to the target hardware. Similar to our work, Zhang et al. (2020) also construct the schedule piece-by-piece. Instead of finding a single (local) mode of the proxy, our work proposes to learn the full distribution over the proxy to improve the robustness against inaccuracies in the proxy.

**Generative Flow Networks.**  GFlowNets have been applied to generating small molecules (Bengio et al., 2021a), Bayesian networks (Deleu et al., 2022), discrete images (Zhang et al., 2022), and biological sequences (Jain et al., 2022). We extend its application to scheduling, a classical combinatorial optimization problem. Similar to previous works, our state-action space is also a DAG, hence training the policy with maximum entropy reinforcement learning methods is inadequate (Bengio et al., 2021a). Our robust scheduling approach shares the same motivation as methods in drug discovery which leverage cheap proxies to generate multiple candidates to be evaluated in the true environment (Bengio et al., 2021a; Jain et al., 2022). Conditional GFlowNets have previously only been theoretically discussed by Bengio et al. (2021b). We enable training conditional GFlowNets with our proposed log-partition variance loss and empirically demonstrate generalization to unseen computation graphs. Note that this differs from previous work that tests the generalization of GFlowNets to unseen data (Nica et al., 2022). To control the selectiveness of the generator, previous works augment the reward with a fixed temperature (Bengio et al., 2021a; Deleu et al., 2022; Jain et al., 2022). Instead, we condition the policy neural network on the temperature term which allows us to tune the selectiveness of the generator at inference time.

## 5  EXPERIMENTS

In this section, we evaluate different aspects of our generative scheduling approach by incrementally adding complexity to the computation graph dataset. First, we restrict training and evaluation to a single computation graph, which corresponds to the same unconditional setting considered by previous works on GFlowNets (Bengio et al., 2021a; Deleu et al., 2022; Jain et al., 2022). Next, we train with multiple computation graphs and evaluate on unseen ones. To the best of our knowledge,

this is the first time that the generalization of conditional GFlowNets to unseen conditioning is tested empirically. Finally, we verify the generalization ability on real-world computation graphs of neural networks that are being used in a diverse set of AI products.

**Experimental setup.** In all experiments, we only use the node time duration as a feature of the computation graph. For simplicity and ease of reproducibility, we avoid any complicated heuristics to add extra features. All our experiments are based on four homogenous devices, which implies that the speedup is upper bounded by 4. In practice, most computation graphs have a lower maximal possible speedup due to their precedence constraints.

**Candidate sampler.** We consider two heuristic and two neural methods for generating candidate schedules. The first is our GFlowNet approach described in Section 3 from which we generate 1000 samples at temperature $\sigma = 0.005$ and take the top 100 following the proxy; the other three are:

- Critical path-based list scheduling, a heuristic algorithm for scheduling on homogeneous devices (Micheli, 1994). List scheduling first forms a topological order of the operations and then assigns them in that order one by one to a free device. In our implementation, we use the Critical Path method (Micheli, 1994) to order the operations. It ensures that operations on the time critical path are scheduled first. This method produces a single schedule.

- Biased Random Key Genetic Algorithm (BRKGA) (Gonçalves & Resende, 2011), a genetic algorithm that has previously shown good performance on scheduling tasks (Paliwal et al., 2020). We use the top 100 schedules from the final population as the candidate schedules.

- Proximal Policy Optimization (PPO) (Schulman et al., 2017), a deep reinforcement learning method that has been successfully applied to scheduling problems (Zhou et al., 2020). PPO also trains a stochastic policy, which makes it a natural choice for comparison with GFlowNets (Bengio et al., 2021a). We employ the same definitions of states, actions, and reward function (with temperature $\sigma = 0.25$; lower was not beneficial) as the GFlowNet approach. To ensure that PPO keeps exploring even after finding a local optimum we employ entropy regularization and decay both the entropy coefficient (Ahmed et al., 2019) and the learning rate to ensure convergence to a good solution. Same as for GFlowNets, we sample 1000 schedules and pick the top 100 as the candidate schedules.

**Metrics.** We measure the performance in terms of the speedup $U(x)$. For the diversity, we report three different measures: graph-edit distance (GED), the L2 distance between the proxy start-times ($d_{\text{inv}}$), and the L2 distance between the proxy start-times concatenated with the device placement ($d_{\text{sen}}$). For diversity, we report the average pairwise distances over the top 100 candidate schedules. See Appendix E.2 for more details on diversity measures.

## 5.1 Proxy errors: diversity for robust scheduling

We examine how differences between the proxy and the target performance model can affect the final runtime. To do so, we first focus on a single computation graph that is used both for training and testing to avoid possible confounding factors that may happen in the generalization setting. Based on the possible reasons for proxy errors discussed in Section 2.2 we design three different target models that each reflect a different setting. In the first setting node durations are incorrectly profiled (Noisy Runtimes). In the second and third settings, the target models the memory movement across devices with a linear model (Valiant, 1990), which can be either bottlenecked by limited bandwidth (Bandwidth Limited), or by high latency (Latency Limited). The linear model has been shown to be a

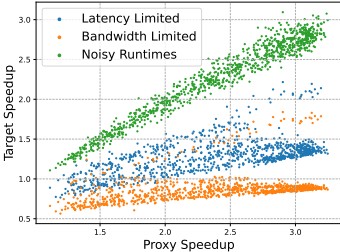

Figure 2: Correlation between proxy and target speedup for different target environments. Modes with varying performance can be observed for a fixed proxy speedup.

Table 1: Robustness results on a single computation graph. We compare different methods for generating candidate schedules. Higher diversity correlates with better robustness against a mismatch of the proxy and the target, with GFlowNet achieving the best diversity and the best target performance on average.

| | Speedup | | | | Diversity | | |
|---|---|---|---|---|---|---|---|
| | Proxy | Noisy Runtimes | Bandwidth Limited | Latency Limited | GED | $d_{inv}$ | $d_{sen}$ |
| List scheduling | 3.23±0.00 | 2.75±0.00 | 1.02±0.00 | 1.74±0.00 | 0 | 0 | 0 |
| BRKGA | 3.22±0.00 | 2.86±0.15 | 1.29±0.45 | 1.80±0.34 | 55.92±2.56 | 22.83±2.39 | 56.21±1.50 |
| PPO | 3.28±0.07 | 3.07±0.09 | 1.38±0.49 | 1.87±0.38 | 85.08±3.54 | 31.71±0.05 | 105.64±0.08 |
| GFlowNet | 3.21±0.02 | 3.05±0.04 | 1.78±0.03 | 2.11±0.03 | 94.79±0.15 | 42.08±0.33 | 115.98±0.09 |

good makespan estimator for certain devices (Hockney, 1994; Culler et al., 1993). We refer to Appendix E.3 for more details. In Figure 2, we show the correlation between the proxy and the different target environments. For all three targets, the proxy is highly correlated but can have target speedups that differ by a factor of up to ×2 for the schedules with high proxy speedups.

We report the speedups and diversity measures in Table 1. The results highlight that any method that can generate multiple good candidate schedules achieves higher speedups on the target environments than list scheduling, which only produces a single unique schedule. Furthermore, if the candidate schedules are more diverse — as is the case for GFlowNet — the target performance is also better on average. PPO and BRKGA exhibit high variability in performance between different runs, where a few runs end up with high speedups on some targets, and other runs result in much lower target speedups. In contrast, the GFlowNet model is consistent over the different random seeds, both in terms of diversity and speedup. The results confirm our hypothesis that a diverse set of candidate schedules with high average proxy speedups can improve robustness towards a misspecified proxy.

## 5.2 GENERALIZING TO UNSEEN COMPUTATION GRAPHS

Next, we evaluate how well our conditional GFlowNet can generalize to unseen computation graphs. We train and evaluate on a diverse set of synthetic computation graphs sampled from different random graph distributions. In particular, we train on graphs of size 50 sampled from the random graph distributions (a) Erdős–Rényi (Erdős et al., 1960), and (b) Layered Graphs (Gagrani et al., 2022) and evaluate, in addition to (a) and (b), on stochastic block model (Holland et al., 1983), Watts-Strogatz (Watts & Strogatz, 1998), and Barabási–Albert (Albert & Barabási, 2002). For details on the generative process of the computation graphs, we refer to Appendix E.5.

In Table 2, we demonstrate that both PPO and the conditional GFlowNet are able to generalize to previously unseen computation graphs, regardless of whether they originate from the same random graph distribution. Next, we ablate our proposed temperature conditioning method by generating 1000 samples at different temperature values. In Figure 3, we observe that decreasing the temperature does indeed shift the sample distribution to the right and also sharpens it when the temperature approaches zero. Notably, the temperature $\sigma = 0.005$ is not in the training distribution, which demonstrates that the model can extrapolate to temperature values outside of the training range. Surprisingly, we observe that training with a variable temperature can improve the performance further than is possible with a fixed temperature, which we demonstrate in Appendix F.

Table 2: Generalization to different random graph distributions. We report the speedup and diversity for the top 100 schedules. PPO and GFlowNet are trained on graphs from the Erdos-Renyi and Layered Graph distribution and we report the average performance over all random graph distributions. The Speedup Proxy 100 column reports the average proxy speedup over the top 100 schedules.

| | Speedup | | Diversity | | |
|---|---|---|---|---|---|
| | Proxy 1 | Proxy 100 | GED | $d_{inv}$ | $d_{sen}$ |
| List scheduling | 3.44 | 3.44 | 0 | 0 | 0 |
| BRKGA | 3.46 | 3.45 | 46.59 | 12.75 | 40.11 |
| PPO | 3.48 | 3.46 | 69.54 | 13.45 | 80.84 |
| GFlowNet | 3.46 | 3.41 | 92.02 | 24.27 | 90.17 |

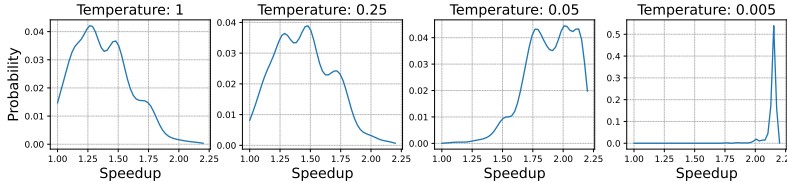

Figure 3: Empirical reward distribution on a 25-node Erdos-Renyi graph for different inference temperatures in the conditional GFlowNet. Lower temperatures allocate more probability mass to better schedules.

## 5.3 REAL WORLD COMPUTATION GRAPHS

Finally, we verify the generalization ability on a small set of real-world computation graphs used for the commercial development of our artificial intelligence hardware and software products (see Appendix E.6 for details). We report the speedup on the same target models used in Section 5.1 to assess robustness on unseen real-world computation graphs. To speed up training, we apply the graph subsampling strategy presented in Section 3.3 to randomly pick between 25 to 75 nodes at every training step.

In Table 3, we observe that the conditional GFlowNet retains the benefits of high diversity and robustness to misspecifications in the proxy even when applied to graphs not seen during training and of larger sizes. PPO shows unstable training behavior and the reward training curve does not converge, despite using the same hyperparameters that worked for the previous two experiments. We conjecture that this is due to the inhomogeneous maximum possible speedup of the training graphs that lead to different reward scales per training graph. In comparison, GFlowNet still converges as before without any changes to the hyperparameters. Note that while PPO exhibits higher diversity than compared to BRKGA, it still underperforms BRKGA due to low average proxy speedups. This highlights that high diversity alone is not sufficient, otherwise, a uniform distribution as the forward policy would already suffice.

We ablate our proposed log-partition variance loss by comparing it against the trajectory balance loss that uses a Topoformer to predict $\log Z$ given a computation graph. Learning such a model is difficult due to large differences in the output space of different computation graphs that arise from the differences in the number of nodes, which in turn impedes the training progress of the policy network. We confirm in Appendix E.6 that our proposed loss function remedies the slow start problem of the baseline and achieves a higher speedup in the end.

Table 3: Generalization on real-world graphs. We train on a small set of real-world graphs and evaluate on unseen ones. GFlowNet retains a high diversity and exhibits consistently better performances than the baselines on the target models. PPO uses the same hyperparameters as in the previous experiments but does not manage to converge on this dataset.

| | Speedup | | | | | Diversity | | |
|---|---|---|---|---|---|---|---|---|
| | Proxy 1 | Proxy 100 | Noisy Runtimes | Bandwidth Limited | Latency Limited | GED | $d_{\text{inv}}$ | $d_{\text{sen}}$ |
| List scheduling | 2.74±0.00 | 2.74±0.00 | 2.51±0.00 | 0.89±0.00 | 1.43±0.00 | 0 | 0 | 0 |
| BRKGA | 2.59±0.18 | 2.58±0.18 | 2.46±0.16 | 1.55±0.17 | 1.80±0.18 | 52.32±21.59 | 17.14±8.17 | 42.64±12.23 |
| PPO | 2.41±0.20 | 2.23±0.27 | 2.28±0.26 | 0.91±0.20 | 1.43±0.10 | 53.05±7.27 | 42.70±3.44 | 64.92±4.08 |
| GFlowNet | 2.71±0.03 | 2.66±0.01 | 2.71±0.01 | 1.73±0.01 | 1.95±0.03 | 87.95±0.13 | 26.56±0.56 | 91.33±0.15 |

## 6 CONCLUSION

We have empirically demonstrated how the conventional optimization approach to scheduling, which optimizes a proxy of the real makespan, is brittle to modeling failures in the proxy itself. Our proposed approach evaluates multiple schedules on the target and thereby achieves more robustness to discrepancies between the proxy and the target. We demonstrated that GFlownets can sample a diverse set of candidate schedules that achieve better target performance than alternative methods which achieve lower diversity. Further, we showed that conditioning on temperature allows a trade-off between diversity and proxy performance, and that conditional GFlowNets can generalize to unseen computation graphs. Interesting future directions include scaling up our method to larger graphs and integrating scheduling heuristics to speed up training.

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

## A  PROOF FOR THEOREM 1

In the following, we will denote $F^R$ to be the flow corresponding to a flow function $F : \mathcal{T} \to \mathbb{R}_{\geq 0}$ with corresponding reward $R$. That is, $F^R : \mathcal{T} \to \mathbb{R}_{\geq 0}$ is a flow function such that for any terminal state $x$ we have $R(x) = \sum_{\boldsymbol{s} \in \mathcal{T}_x} F^R(\boldsymbol{s})$, where $\sum_{\boldsymbol{s} \in \mathcal{T}_x} F^R(\boldsymbol{s}) =: F^R(x)$ is the total flow at the terminal state $x$.

**Theorem** (Flow Continuity). *Let $\{R_i\}_{i=1}^{\infty}$ be a sequence of nonnegative reward functions such that for all terminal states $x$, $R_i(x) \to R(x)$ as $i \to \infty$. Then, for any flow $F^R$ with reward $R$, there exists a sequence of flow functions $\{F^{R_i}\}_{i=1}^{\infty}$ with $F^{R_i}(\boldsymbol{s}) \to F^R(\boldsymbol{s})$ for all $\boldsymbol{s} \in \mathcal{T}$.*

*Proof.* Let $\{x_i\}_{i=1}^{M}$ be the set of terminal states and define

$$F^{R_i}(\boldsymbol{s} = (s_0, \dots, x)) := \begin{cases} F^R(\boldsymbol{s}) \frac{R_i(x)}{R(x)} & \text{if } R(x) > 0, \\ \frac{R_i(x)}{|\mathcal{T}_x|} & \text{else.} \end{cases} \tag{10}$$

To see that $F^{R_i}$ is a valid flow for $R_i$, we note that $F^{R_i} \geq 0$ and for any terminal state $x$ with $R(x) > 0$ we have

$$\begin{aligned} F^{R_i}(x) = \sum_{\boldsymbol{s} \in \mathcal{T}_x} F^{R_i}(\boldsymbol{s}) &= \frac{R_i(x)}{R(x)} \sum_{\boldsymbol{s} \in \mathcal{T}_x} F^R(\boldsymbol{s}) \\ &= \frac{R_i(x)}{R(x)} R(x) = R_i(x). \end{aligned} \tag{11}$$

Using similar reasoning we arrive at $F^{R_i}(x) = R_i(x)$ when $R(x) = 0$. Finally, for any $\boldsymbol{s} \in \mathcal{T}$ with terminal state $x$ and $R(x) > 0$ we have

$$F^{R_i}(\boldsymbol{s}) = \frac{R_i(x)}{R(x)} F^R(\boldsymbol{s}) \to F^R(\boldsymbol{s}). \tag{12}$$

In the case of $R(x) = 0$, we note that $F^R(\boldsymbol{s}) = 0$ and

$$F^{R_i}(\boldsymbol{s}) = \frac{R_i(x)}{|\mathcal{T}_x|} \to \frac{R(x)}{|\mathcal{T}_x|} = 0, \tag{13}$$

thus proving convergence of $F^{R_i}$ to $F^R$. ∎

**Corollary 1.** *Let $\log R_{\sigma_i} := \log R(x; m, \sigma_i) = (U(x) - m)/\sigma_i$ be a sequence of temperature conditioned log reward functions with $\sigma_i \searrow \sigma_0$. Then, for any $\epsilon > 0$ and flow $F^{R_{\sigma_0}}$ there exists a neighborhood $(\sigma_0, \sigma_0 + \delta)$ containing flow functions $F^{R_\sigma}$ with $F^{R_\sigma}(\boldsymbol{s}) - F^{R_{\sigma_0}}(\boldsymbol{s}) < \epsilon$ for all $\boldsymbol{s} \in \mathcal{T}$. Furthermore, $F^{R_{\sigma_i}}(\boldsymbol{s})$ monotonically decreases to $F^{R_{\sigma_0}}(\boldsymbol{s})$ for all $\boldsymbol{s} \in \mathcal{T}$ as $i$ goes to $0$.*

The above corollary suggests that it is feasible to use a single neural network — that can approximate arbitrary *continuous* functions — to learn all flow functions for different temperature values.

## B  FAILURE MODES WHEN ESTIMATING THE LOG-PARTITION FUNCTION

In this section, we present the different cases in which an incorrect estimate of $\log Z$ can lead to gradients that point in the opposite direction of the "true gradients". We start by rewriting the trajectory balance loss (Malkin et al., 2022) as:

$$\begin{aligned} \mathcal{L}_{\text{TB}}(\boldsymbol{s}; \boldsymbol{\theta}) = \frac{1}{2} \bigg( & \log Z(G_C; \boldsymbol{\theta}_Z) + \sum_{t=1}^{n} \log P_F(s_t | s_{t-1}; \boldsymbol{\theta}_P) \\ & - \log R(x) - \sum_{t=1}^{n} \log P_B(s_{t-1} | s_t; \boldsymbol{\theta}_P) \bigg)^2 \end{aligned} \tag{14}$$

For better readability, we do not explicitly write out the dependence on the computation graph $G_C$ for the trajectories, their probabilities, and the reward. The parameters of the neural networks are denoted as $\boldsymbol{\theta}_P$ and $\boldsymbol{\theta}_Z$ respectively. We can rearrange the terms in Equation 14, such that:

$$
\mathcal{L}_{\mathrm{TB}}(\boldsymbol{s}; \boldsymbol{\theta}) = \frac{1}{2}\Bigg(\underbrace{\sum_{t=1}^{n}\log P_F(s_t|s_{t-1}; \boldsymbol{\theta}_P)}_{\varphi_{\mathrm{F}}} \\
-\underbrace{\left(\log R(x) + \sum_{t=1}^{n}\log P_B(s_{t-1}|s_t; \boldsymbol{\theta}_P) - \log Z(G_C; \boldsymbol{\theta}_Z)\right)}_{\varphi_{\mathrm{RBZ}}}\Bigg)^2
\tag{15}
$$

In words, $\varphi_{\mathrm{F}}$ is the log probability of the trajectory $\boldsymbol{s}$ as computed by the forward distribution, and $\varphi_{\mathrm{RBZ}}$ can also be viewed as the log probability of $\boldsymbol{s}$ but instead estimated by a combination of the reward, log-partition function, and the backward distribution. To simplify the argument, we assume that the backward distribution is independent of $\boldsymbol{\theta}$, which can be realized for example by setting it equal to the distribution that assigns equal probability to all parents. Thus, we denote the true log-partition function as $\log Z^*(G_C)$ and by assuming access to it when computing the loss we can rewrite part of Equation 15 as:

$$
\varphi_{\mathrm{RBZ}^*} = \log R(x) + \sum_{t=1}^{n}\log P_B(s_{t-1}|s_t) - \log Z^*(G_C)
\tag{16}
$$

We denote the loss using $\varphi_{\mathrm{RBZ}^*}$ as $\mathcal{L}_{\mathrm{TB}}^*(\boldsymbol{s}; \boldsymbol{\theta})$.

There exist two cases in which $\mathcal{L}_{\mathrm{TB}}^*(\boldsymbol{s}; \boldsymbol{\theta})$ can be non-zero and in both cases, inaccurate estimation of the log-partition function can lead to wrong gradients for the policy neural network. In the first case, we assume that the neural network that estimates $P_F$ is overestimating the probability of $\boldsymbol{s}$:

$$
\varphi_{\mathrm{F}} - \varphi_{\mathrm{RBZ}^*} > 0.
\tag{17}
$$

Furthermore, we consider the scenario in which the regression model makes an error when predicting the log-partition value. In particular, we assume that the neural network is underestimating the true value by an amount that is greater than the error $\varphi_{\mathrm{F}} - \varphi_{\mathrm{RBZ}^*}$, i.e., we assume:

$$
\varphi_{\mathrm{F}} - \varphi_{\mathrm{RBZ}^*} < \log Z^*(G_C) - \log Z(G_C; \boldsymbol{\theta}_Z)
\tag{18}
$$

The implications of this are:

$$
\varphi_{\mathrm{F}} - \varphi_{\mathrm{RBZ}^*} < \log Z^*(G_C) - \log Z(G_C; \boldsymbol{\theta}_Z)
\tag{19}
$$
$$
\Rightarrow \qquad \varphi_{\mathrm{F}} < \log Z^*(G_C) - \log Z(G_C; \boldsymbol{\theta}_Z) + \varphi_{\mathrm{RBZ}^*}
\tag{20}
$$
$$
\Rightarrow \qquad \varphi_{\mathrm{F}} < \varphi_{\mathrm{RBZ}}
\tag{21}
$$
$$
\Rightarrow \qquad \varphi_{\mathrm{F}} - \varphi_{\mathrm{RBZ}} < 0
\tag{22}
$$

Equation 17 and Equation 22 imply that $-\nabla_{\boldsymbol{\theta}_P}\mathcal{L}_{\mathrm{TB}}(\boldsymbol{s}; \boldsymbol{\theta})$ is an ascent direction for $\mathcal{L}_{\mathrm{TB}}^*(\boldsymbol{s}; \boldsymbol{\theta})$, instead of the desired descent direction. In detail, for non-zero gradients the inner product is positive:

$$
\langle -\nabla_{\boldsymbol{\theta}_P}\mathcal{L}_{\mathrm{TB}}(\boldsymbol{s}; \boldsymbol{\theta}), \nabla_{\boldsymbol{\theta}_P}\mathcal{L}_{\mathrm{TB}}^*(\boldsymbol{s}; \boldsymbol{\theta})\rangle
\tag{23}
$$
$$
= \langle -(\varphi_{\mathrm{F}} - \varphi_{\mathrm{RBZ}})\nabla_{\boldsymbol{\theta}_P}\varphi_{\mathrm{F}}, (\varphi_{\mathrm{F}} - \varphi_{\mathrm{RBZ}^*})\nabla_{\boldsymbol{\theta}_P}\varphi_{\mathrm{F}}\rangle
\tag{24}
$$
$$
> 0
\tag{25}
$$

In the second case, $\varphi_{\mathrm{F}} - \varphi_{\mathrm{RBZ}^*} < 0$, we assume that the neural network overestimates the log-partition function. Analog to the first case we can show that the gradients point in the opposite direction of the true gradients.

These two cases happen more frequently when $\log Z(G_C; \boldsymbol{\theta}_Z)$ is more challenging to regress accurately. For example, when there are computation graphs with drastically different node numbers, their $\log Z(G_C; \boldsymbol{\theta}_Z)$ values can have significant differences (e.g., 10s vs. 1000s). Training a regression model on such large ranges is notoriously difficult (Pohlen et al., 2018). Our proposed log-partition variance loss in Equation 8 avoids this regression task completely, and we confirm its benefits empirically in Appendix E.6.

## C  GFLOWNET TRAINING PROCESS

In each parameter update step, we first sample a single computation graph from the training dataset, next we sample a single reward temperature from the log-uniform distribution, then we sample $b$ trajectories — we refer to these as a mini-batch — for that computation graph and reward temperature, and finally, we compute the loss based on the mini-batch of trajectories. The trajectories are sampled using the current forward policy $P_F$ conditioned on the temperature. Empirically we observe that the temperature conditioning method allows us to forgo using a special exploratory policy that mixes $P_F$ and a uniform distribution over the allowed actions, which was used by previous works.

## D  TEMPERATURE-CONDITIONED TOPOFORMER

The Topoformer (Gagrani et al., 2022) has the same structure as the Transformer (Vaswani et al., 2017) encoder, that is it stacks $L$ layers and each layer $l$ (for $1 \le l \le L$) consists of two steps:

$$\hat{h}^{(l)} = h^{(l-1)} + \text{MHA}_{\text{Topoformer}}(\texttt{LayerNorm}(h^{(l-1)})) \tag{26}$$

$$h^{(l)} = \hat{h}^{(l)} + \texttt{MLP}(\texttt{LayerNorm}(\hat{h}^{(l)})) \tag{27}$$

where $\text{MHA}_{\text{Topoformer}}$ denotes the Topoformer version of the multi-head attention, and $h^{(0)}$ is the input to the Topoformer, which is the output of a linear layer (i.e., an affine transformation with no activation function) applied on the computation graph and state features. Topoformer uses the same $\texttt{MLP}$ as in the original Transformer:

$$\texttt{MLP}(\hat{h}^{(l)}) = \texttt{lin}_2(\texttt{ReLU}(\texttt{lin}_1(\hat{h}^{(l)}))) \tag{28}$$

We inject the temperature $\sigma$ by replacing $\texttt{lin}_1(\hat{h}^{(l)})$ with $\texttt{lin}_1(\hat{h}^{(l)}, e_\sigma)$:

$$e_\sigma = \texttt{ReLU}(\texttt{lin}_b(\texttt{ReLU}(\texttt{lin}_a(\sigma)))) \tag{29}$$

$$\texttt{lin}_1(\hat{h}^{(l)}, e_\sigma) = \texttt{lin}_{\text{scale}}(e_\sigma) \odot \texttt{lin}_1(\hat{h}^{(l)}) + \texttt{lin}_{\text{shift}}(e_\sigma) \tag{30}$$

Note that $e_\sigma$ is the same for all layers $l$ and Equation 30 corresponds to Equation 9 in the main paper.

## E  EXPERIMENT DETAILS

### E.1  CANDIDATE SAMPLERS

We use the popular open-source library pymoo (Blank & Deb, 2020) to implement the BRKGA candidate sampler. Our PPO implementation is based on algorithm 1 at https://spinningup.openai.com/en/latest/algorithms/ppo.html, and we follow Schulman et al. (2017) to implement the entropy regularisation by adding the entropy term directly to the PPO-clip loss. We decay this entropy term during training similar to Ahmed et al. (2019). We use the same learning rate for both the actor and the critic, and we decay it with an exponential schedule.

We train GFlowNets conditioned on a temperature randomly sampled between 0.01 and 1. At inference, we use 0.005 for the temperature in all experiments. We use the Adam optimizer with default hyperparameters to optimize the parameters. We compute the gradients at each update step based on a minibatch that consists of 100 sampled trajectories for a single computation graph and use a single temperature value to compute their rewards. We observed no benefits in our initial experiments that used multiple computation graphs or temperatures in a single minibatch.

We represent the computation graph as a directed graph with a single node feature (runtime of the operation). In addition, we represent different states by concatenating a state feature vector to each node, which consists of:

- Start time assigned by the proxy (default: $-1$)
- Device placement as a one-hot vector (default: 0 vector)
- Binary indicator: 1 if adding it to the schedule is a valid action, else 0
- Binary indicator: 1 if removing it is a valid backward-action, else 0

- Binary indicator: 1 if the node is already part of the schedule, else 0

The combined computation graph and state features are treated as a single graph by the Topoformer neural network.

### E.2 METRICS

The graph-edit distance (GED) compares two schedules in their chain-graphs form. In particular, we can model a schedule for a computation graph, by constructing a chain graph for each device that specifies the additional precedence constraints we introduce to complete the order in which the operations are run on each device. The GED is then computed simply by taking the difference between the adjacency matrices and normalizing it by the total number of edges.

The L2 distance between the start ($d_{inv}$) times simply takes the start times as assigned by the proxy model and computes the L2 norm of the difference.

The L2 distance including the device assignment ($d_{sen}$) additionally concatenates the device placement to the times.

### E.3 TARGET ENVIRONMENT WITH LINEAR MEMORY MODEL

The linear memory model (Valiant, 1990) computes the delay as a linear function $f(m) = am + b$ of the memory $m$ with $a$ modeling the amount of data that can be transferred per time and $b$ modeling the startup delay. In the Bandwidth Limited setting the $a$ term dominates the delay, while in the Latency Limited setting $b$ has a greater effect.

### E.4 PROXY ERRORS: DIVERSITY FOR ROBUST SCHEDULING

In this experiment, we consider a single real-world graph that has around 78 nodes.

### E.5 GENERALIZATION TO UNSEEN COMPUTATION GRAPHS

We generate the synthetic graph dataset from random graph distributions over undirected graphs. To get DAGs from these graphs, we randomly choose a direction for every edge in a way that produces no cycles. We closely follow the setup described in Appendix A.1.4 by Paliwal et al. (2020), and the setup described in Appendix A by Gagrani et al. (2022).

We sample the runtimes for each node from the uniform distribution $\mathcal{U}(0, 1)$.

For training, we use 1000 different computation graphs, with equally many sampled from the two random graph distributions: Erdős–Rényi (Erdős et al., 1960), and Layered Graphs (Gagrani et al., 2022). We report test performances on 50 different computation graphs with equally many sampled from the five different random graph distributions: Erdős–Rényi (Erdős et al., 1960), Layered Graphs (Gagrani et al., 2022), stochastic block model (Holland et al., 1983), Watts-Strogatz (Watts & Strogatz, 1998), and Barabási–Albert (Albert & Barabási, 2002).

### E.6 REAL-WORLD COMPUTATION GRAPHS

The computation graphs in this dataset originate from a diverse set of neural network architectures with different applications, including for example classification and denoising. We train on 8 real-world computation graphs of sizes below 100 nodes and evaluate on 4 different computation graphs of sizes between a dozen and 128 nodes. Note that the maximal achievable speedup is on average lower for the real-world computation graphs compared to the synthetic ones. Furthermore, the synthetic graphs are also more homogenous in terms of the maximal achievable speedup, with a lower limit of at least 3. In contrast, some real-world graphs could not exceed a speedup of 1.5 and others went beyond 3.

In Figure 4, we report the average speedup of the schedules sampled during training, at varying training steps. We compare our proposed log-partition variance loss against the trajectory balance loss (Malkin et al., 2022) that uses a Topoformer (Gagrani et al., 2022) to regress the log-partition

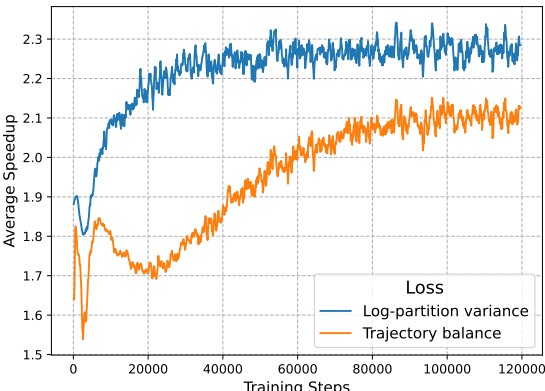

Figure 4: Average speedup of schedules sampled during training. We compare the log-partition variance loss against the trajectory balance loss that models the log-partition function with a neural network. The log-partition variance loss starts improving earlier and achieves a much higher final speedup.

function. The results demonstrate that the log-partition variance loss starts to improve the average speedup much earlier on during training and achieves a better performance in the end.

## F  TEMPERATURE CONDITIONING ABLATION

In order to increase the likelihood of sampling good schedules, one could introduce a fixed temperature throughout training and inference. However, we have observed that this procedure is unreliable for small temperatures. Figure 5 shows generalization performance during training on the synthetic graph experiment of Section 5.2. As can be seen, choosing a temperature of $0.01$ results in a smaller maximum reward as opposed to training on a higher temperature of $0.03$. On the other hand, sampling a range of temperatures between $0.01$ and $1$ and evaluating on $0.01$ samples the best performing schedules on unseen computation graphs.

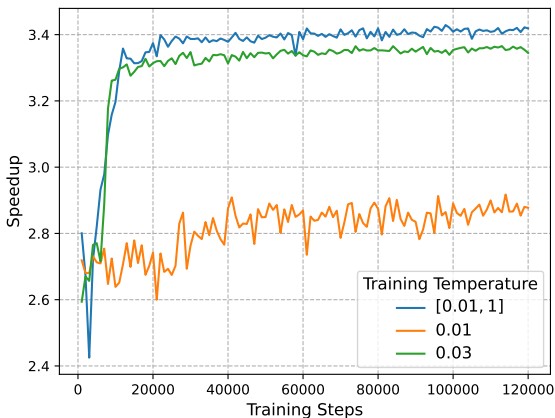

Figure 5: The impact of different temperature regimes on top-1 generalization performance. Training on single temperatures prevents learning when set too low (orange). On the other hand, training on a range of different temperatures (blue) results in better performance when performing inference with the minimum training temperature.

