# OpenReview forum: "Robust Scheduling with GFlowNets"
_ICLR.cc/2023/Conference — ICLR 2023 poster_

### Official Review · Reviewer_9S3Z · 2022-10-22

**Confidence:** 5
**Correctness:** 3
**Technical Novelty And Significance:** 3
**Empirical Novelty And Significance:** 4
**Recommendation:** 6

**Clarity, Quality, Novelty And Reproducibility:**

The paper is clearly written, of good quality. Novelty is mostly in the form of experiments and a new application domain for a fairly new method (introduced at NeurIPS 2021).


**Strength And Weaknesses:**

GFlowNets are a new RL variant (less than a year old) with only a few applications yet, so this paper introduces a new type of application, in the realm of scheduling, and an approach which may be applicable more broadly. The successfully show empirical success on two GFlowNet novelties: conditioning on a rich input (the computational graph) and on temperature (to control the diversity-reward trade-off).

In page 5, they claim that previous works on GFlowNets did not consider the conditional case, but this is not true and they themselves mention in section 4 that Bengio et al 2021b introduced the theoretical framework for conditional GFlowNets. What is true is that may be the first (or among the first, with concurrent work) to validate experimentally a conditional GFlowNet and show its usefulness.



**Summary Of The Paper:**

The authors apply GFlowNets to the problem of scheduling operations in a computational graph on homogeneous parallel hardware. The evaluate conditional GFlowNets for the first time with the computational graph being the conditioning input and the schedule being the output. They also innovate by conditioning on the reward temperature in order to find a good trade-off between diversity of the generated samples and their average reward. Diversity is important in this application because the generative policy is trained using an imperfect but cheap to compute proxy and only a few of the generated schedules are then evaluated on the ground truth hardware. GFlowNets are interesting in this context because their training objective makes them sample with probability proportional to the reward (rather than maximizing the reward as in other RL approaches). Experiments show improvements against previous methods, including standard RL methods.

**Summary Of The Review:**

This paper introduced a new form of application for GFlowNets, mapping a computational graph to a distribution over efficient schedules for it. To achieve this, the demonstrate the feasibility and success of the conditional GFlowNet formalism and also show how to use it to control the trade-off between diversity and average reward by conditioning on the reward temperature (exponent of the reward).
I consider the paper should be published if they fix their claim about conditional GFlowNets (that it was not considered before, see above).

---

> ### Author Response · Authors · 2022-11-16
> **Response to Reviewer 9S3Z**
>
> We sincerely thank Reviewer 9S3Z for their positive feedback and important comment. The claim on page 5 was indeed incorrect in the way it was phrased and we removed it in the updated paper.

---

### Official Review · Reviewer_uJuK · 2022-10-24

**Confidence:** 4
**Correctness:** 3
**Technical Novelty And Significance:** 3
**Empirical Novelty And Significance:** 3
**Recommendation:** 8

**Clarity, Quality, Novelty And Reproducibility:**

The paper is fairly clear, although there are many details lacking. I am not sure I could exactly reproduce the results of this paper as-is, although probably a fair approximation of it. The paper feels fairly minimal on the empirical side, it has some minimal experiments which are interesting, but it wouldn't hurt for it to be more thorough.

In terms of novelty, the proposed changes are reasonable ways to improve GFlowNets, but as far as I can tell are not groundbreaking ideas.

**Strength And Weaknesses:**

**Strengths**:
- clear writing, clear exposition, clear methods
- brings novel insights into GFlowNet research
- good use of the diversity-as-a-strength feature of GFlowNets to improve scheduling

**Weaknesses**:
- there are some details lacking, for example, I'm not sure what the architecture is to condition on a particular computation graph, or how temperature is fed to the MLP producing $e_\sigma$.
- there are some convincing results for the temperature-conditional GFN method proposed, but not for the log-partition variance loss. The authors claim that learning $Z_\theta(G_C)$ is divergent, but no evidence is provided to help understand the reader why this is the case, nor if the learned flows converge to the correct quantities.
- similarly, the effect of training on subgraphs vs full graphs is not very deeply discussed nor empirically justified.

> To the best of our knowledge, this is the first time that generalization of GFlowNets is tested empirically.

If I understand prior work correctly, generalization to unseen _data_ has been tested (see e.g. [1]). Perhaps here the authors are instead referring to generalization to unseen conditionals (in the present case, unseen computation graphs to be scheduled)?

> In practice, we use the training distribution to estimate $\mathbb{E}_s [\zeta(s)]$ with a mini-batch of trajectories

How does this interface with (temperature and graph) conditioning? $Z$ is effectively a function of $G_C$ and $\sigma$. If I understand correctly this means that for a given $G_C$ (and $\sigma$) many samples are taken. Does this require a "combinatorial" number of samples? i.e. for a "minibatch" of size $n$ do we need $n=n_b n_G n_\sigma$ samples?

[1] Evaluating Generalization in GFlowNets for Molecule Design, Andrei Cristian Nica, Moksh Jain, Emmanuel Bengio, Cheng-Hao Liu, Maksym Korablyov, Michael M. Bronstein, Yoshua Bengio, MLDD 2022

**Summary Of The Paper:**

This paper examines the computation scheduling problem, recently tackled by RL methods, and attempts to tackle it with the GFlowNet framework. The application is fairly straightforward, but the authors introduce additional interesting aspects that improve performance.

This paper has several contributions:
- applying GFlowNets to the scheduling problem
- training temperature-conditional GFNs (with temperatures sampled log-uniformly)
- training Trajectory Balance-style GFNs without having to estimate $Z(G_C)$ using a variance loss trick
- training on subgraphs as a data augmentation

**Summary Of The Review:**

While I am unfortunately unable to judge the impact of this work on scheduling problems, the proposed improvements to the GFlowNet framework seem interesting. I think the paper could do much more in terms of details, empirical validation of the proposed ideas, and potentially tackle larger scale benchmarks.

---

> ### Author Response · Authors · 2022-11-16
> **Response to Reviewer uJuK**
>
> We sincerely thank Reviewer uJuK for their positive feedback and valuable suggestions. Below we address every comment in detail.
>
> > I'm not sure what the architecture is to condition on a particular computation graph, or how temperature is fed to the MLP producing $e_\sigma$.
>
> We added a more detailed description of what the Topoformer takes as input to Appendix C.1. In summary, the state associated with a (partial) schedule is represented using per-node features that are concatenated to the feature of the computation graph. This graph with extended node features is then used as input to the Topoformer, i.e., it treats the computation graph and the state as a single graph.
>
> > The authors claim that learning $Z_\theta(G_C)$ is divergent, but no evidence is provided to help understand the reader why this is the case, nor if the learned flows converge to the correct quantities.
>
> We updated Section 3.2 to clarify:
> *In order to apply the trajectory balance loss in the conditional case, we would need to learn an additional regression model that estimates the log-partition function $\log Z$ conditioned on $G_C$. Training such a network accurately is difficult but important for learning the probabilities $P_F$. In particular, a wrong estimation of $\log{Z}$ can incorrectly change the direction of the gradients of the loss function. We explain why this occurs in Appendix B.*
>
> Additionally, we add a more formal argument in Appendix B. In summary:
> We observed two issues when learning $\log Z$. First, when there was a significant discrepancy between the initial $\log Z$ prediction and the target $\log Z^*$, then we needed more training steps to reach the same performance. Albeit, this can be partially mitigated by setting a larger learning rate for the parameters that affect the $\log Z$ prediction. Secondly, the main issue is that different computation graphs can have very different $\log Z$ values. This is especially pronounced when we compare two computation graphs with different numbers of nodes — even at reward temperatures close to 0 and even more so at larger values. Predicting an incorrect value for $\log Z$ is a big issue because underestimating it can lead to some trajectories’ probabilities being incorrectly reduced while overestimating it can incorrectly increase the probabilities of some trajectories.
>
> Furthermore, we run an ablation on the real-world computation graph dataset that trains on computation graphs of varying node numbers. In Figure 4 in the Appendix, we see that the model trained with the log-partition variance loss performs much better than the baseline that predicts the $\log Z$ value with a neural network.
>
> > the effect of training on subgraphs vs full graphs is not very deeply discussed nor empirically justified.
>
> The primary issue we intended on addressing with training on random subgraphs is the training efficiency, especially in the case of real-world computation graphs where we cannot simply synthesize more computation graphs of a specific size. We currently sample subgraphs by uniformly picking a subset of nodes. While this can lead to subgraphs with disconnected nodes (which do not occur in the original graphs), we do not observe any problems arising from it. We believe that further investigation into different sampling methods and how this functions as a type of data augmentation for tasks operating on computation graphs is an interesting future avenue.
>
> > If I understand prior work correctly, generalization to unseen data has been tested (see e.g. [1]). Perhaps here the authors are instead referring to generalization to unseen conditionals (in the present case, unseen computation graphs to be scheduled)?
>
> Indeed, we are referring to unseen computation graphs. We update the text as follows:
> *To the best of our knowledge, this is the first time that the generalization of conditional GFlowNets to unseen conditioning is tested empirically.*
>
> In addition, we add the following clarification to Section 4:
> *Note that this differs from previous work that tests the generalization of GFlowNets to unseen data (Nica et al., 2022).*
>
> > … for a "minibatch" of size n do we need n=nbnGnσ samples?
>
> We add the following to Appendix C.1 to clarify:
> *We compute the gradients at each update step based on a minibatch that consists of 100 sampled trajectories for a single computation graph and use a single temperature value to compute their rewards. We observed no benefits in our initial experiments that used multiple computation graphs or temperatures in a single minibatch.*

---

### Official Review · Reviewer_umWR · 2022-10-25

**Confidence:** 4
**Correctness:** 4
**Technical Novelty And Significance:** 3
**Empirical Novelty And Significance:** 3
**Recommendation:** 8

**Clarity, Quality, Novelty And Reproducibility:**

The paper is well-written with few typos or ambiguities.

One of the claimed contributions is the introduction of an alternative to pure reward maximization which is more robust to proxy errors. This might be true for the problem of operation scheduling. It is, however, an established method for other domains, such as drug discovery, where a cheap proxy is used in place of the true reward. In fact, one of the references, Bengio et al., 2021a, uses the exact same rationale in the context of molecule generation. It seems appropriate to point out how this key underlying idea is applied elsewhere.

**Strength And Weaknesses:**

This paper applies the right tool, GFlowNets, to an important problem, operation scheduling, and obtains good results. Note that it might be worth mentioning that the proposed state space is a DAG, which necessitates the GFlowNet machinery; otherwise, max-entropy RL suffices.

Along the way, it also empirically evaluates the idea of conditional GFlowNets and utilizes a log-partition variance loss, which are not done in prior work.

The presented empirical evidence is comprehensive and includes speedup on target hardware and real-world computation graphs.

Approximating E_s[\zeta(s)] with the mini-batch average of \zeta(s) reminds me of variance reduction via baseline subtraction where the baseline is estimated with the batch reward. I encourage the authors to explore how the proposed loss is related to established variance reduction methods in RL and beyond.


**Summary Of The Paper:**

Scheduling operations in a computation graph on parallel hardware is NP-hard. This makes ML approaches such as RL attractive. However, the high cost of evaluation necessitates the use of proxy reward models, which can be overfitted to with reward maximization, resulting in poor performance on actual hardware. This is analogous to drug discovery where a proxy model is used in place of wet-lab experiments.

The solution is to seek not just the best mode in the reward distribution but all modes proportional to their goodness. This can be done by training an amortized sampler which samples proportional to the Boltzmann distribution defined by the EBM (the proxy model). The GFlowNet framework is used to train such a sampler with temperature conditioning. Empirical results show that GFlowNets achieve a higher degree of diversity, which correlates with better on-hardware performance.


**Summary Of The Review:**

Well-motivated paper with adequate theoretical and empirical contributions.

---

> ### Author Response · Authors · 2022-11-16
> **Response to Reviewer umWR**
>
> We sincerely thank Reviewer umWR for their positive feedback and insightful comments. Below we give detailed responses to each comment.
>
> > Note that it might be worth mentioning that the proposed state space is a DAG, which necessitates the GFlowNet machinery; otherwise, max-entropy RL suffices.
>
> Thanks for the suggestion, we agree that it is worth mentioning and added the following to Section 4:
> *Similar to previous works, our state-action space is also a DAG, hence training the policy with maximum entropy reinforcement learning methods is inadequate (Bengio et al., 2021a).*
>
>
> > Approximating $E_s[\zeta(s)]$ with the mini-batch average of $\zeta(s)$ reminds me of variance reduction via baseline subtraction where the baseline is estimated with the batch reward.
>
> Thanks for the interesting pointer.
>
> > One of the claimed contributions is the introduction of an alternative to pure reward maximization which is more robust to proxy errors. This might be true for the problem of operation scheduling. It is, however, an established method for other domains, such as drug discovery, where a cheap proxy is used in place of the true reward.
>
> We agree that the parallels to drug discovery are worth pointing out and added the following to Section 4:
> *Our robust scheduling approach shares the same motivation as methods in drug discovery which leverage cheap proxies to generate multiple candidates to be evaluated in the true environment (Bengio et al., 2021a; Jain et al., 2022).*

---

### Official Review · Reviewer_b75p · 2022-10-30

**Confidence:** 5
**Correctness:** 4
**Technical Novelty And Significance:** 3
**Empirical Novelty And Significance:** 3
**Recommendation:** 8

**Clarity, Quality, Novelty And Reproducibility:**

Paper is very well written, clearly explains the problem and its key ideas.
Its experimentations are also very well explained.

**Strength And Weaknesses:**

+ High impact problem, considering that scheduling is a very generic problem in computer systems.
+ Very solid contribution on how to use GFlowNets in scheduling
+ Solid experiments with good baselines (BRKGA, List Scheduling, GFlowNet)

- Seems to be only simulation (would be great if we can see some real system experiments)


**Summary Of The Paper:**

The paper works on scheduling of a computational graph which is an NP-hard problem.
The paper considers a problem of scheduling a computational graph on a fixed no. of homogeneous devices.
The paper claims that the previous approaches take large number of evaluations for convergence and suggests proxies as a faster alternative.
Instead of trying to minimize the makespan measure w.r.t the proxy, this paper creates a set of candidates and learns a generative model to assign higher probability to the low-makespan schedules.
Overall, the paper is a case study of GFlowNets on scheduling computational graphs.
Main difference over GFlowNet baseline is (1) using a variance loss on the log-partition function, (2) learning a single model for multiple different reward functions by conditioning the policy network on a temperature.



**Summary Of The Review:**

The paper works on scheduling of a computational graph which is an NP-hard problem.
The paper considers a problem of scheduling a computational graph on a fixed no. of homogeneous devices.
The paper combines GFlowNet with Topoformer with appropriate changes to the formulation.

While it may look incremental from a cursory look, the paper makes a very solid contribution.
In fact, its experiments also support its significant benefits.

However, it seems that the results are limited to simulation.
It would be great if we can see more real system experiments.

---

> ### Author Response · Authors · 2022-11-16
> **Response to Reviewer b75p**
>
> ​​We sincerely thank Reviewer B75p for their positive feedback and for the recognition of scheduling as a high impact problem.
>
> We agree that testing our method on real systems is very interesting, and plan on doing so in future work. For the current work, we have included additional references that indicate that the target environments are realistic in some cases and added more explanations on the linear memory model. The updated text in Section 5.1 now reads as:
> *In the second and third settings, the target models the memory movement across devices with a linear model (Valiant, 1990), which can be either bottlenecked by limited bandwidth (Bandwidth Limited), or by high latency (Latency Limited). The linear model has been shown to be a good makespan estimator for certain devices (Hockney, 1994; Culler et al., 1993). We refer to C.3 for more details.*

---

### Author Response · Authors · 2022-11-16
**General response**

We sincerely thank all the reviewers and the area chair for their time reviewing our paper.

Below we would like to highlight some new results and changes to the paper:
- Added an extensive explanation to Appendix B that motivates our proposed log-partition variance loss for the conditional GFlowNet setting
- Added an ablation to Section 5.3 and Appendix C.6 for the log-partition variance loss that provides empirical support to our claims
- Added more experimental details to Appendix C that improve the reproducibility
- Fixed some incorrect expressions and typos

---

### Decision · Program_Chairs · 2023-01-20

**Decision:**

Accept: poster

**Justification For Why Not Higher Score:**

The experiments are somewhat limited and there is no theory supporting the claims.

**Justification For Why Not Lower Score:**

The reviewers liked the paper which presents a novel solution to an important practical problem.

**Metareview: Summary, Strengths And Weaknesses:**

The paper proposes to solve scheduling problems using the recently introduced GFlowNets. Experimental results demonstrate that the method is robust to modeling errors in the proxy metric and can generalize to unseen scheduling problems, somewhat better than earlier ML-based methods in the literature.

**Note From Pc:**

if the above contains the word "oral" or "spotlight" please see: "oral" presentation means -> notable-top-5% and "spotlight" means -> notable-top-25%. As stated in our emails, we are disassociating presentation type from AC recommendations